# More than Votes? Voting and Language based Partisanship in the US Supreme Court

**Biaoyan Fang**◇,♣  **Trevor Cohn**◇*  **Timothy Baldwin**◇,♠  **Lea Frermann**◇

◇The University of Melbourne, Australia
♠MBZUAI  ♣CSIRO Data61

{biaoyan,tcohn,tbaldwin,lfrermann}@unimelb.edu.au

## Abstract

Understanding the prevalence and dynamics of justice partisanship and ideology in the US Supreme Court is critical in studying jurisdiction. Most research quantifies partisanship based on voting behavior, and oral arguments in the courtroom — the last essential procedure before the final case outcome — have not been well studied for this purpose. To address this gap, we present a framework for analyzing the language of justices in the courtroom for partisan signals, and study how partisanship in speech aligns with voting patterns. Our results show that the affiliated party of justices can be predicted reliably from their oral contributions. We further show a strong correlation between language partisanship and voting ideology.[1]

## 1 Introduction

The study of partisanship and ideology has been an important topic in understanding the US legal system (Jacobi and Sag, 2018; Devins and Baum, 2017; Bonica and Sen, 2021; Doerfler and Moyn, 2021). Most research has focused on justice voting patterns (Bonica and Sen, 2021; Martin and Quinn, 2002, 2007; Epstein et al., 2007a; Bailey, 2013) and behavior in court, e.g. the frequency of interruptions (Epstein et al., 2010) or questions (Epstein and Weinshall, 2021). Despite their core role in legal determination, the *content* in terms of whether oral arguments portray partisan values has received less attention (Bergam et al., 2022).

In political discourse in particular, word choice is nuanced to convey specific messages (Lakoff, 2010; Jarvis, 2004; Robinson et al., 2017; Jensen et al., 2012; Dutta et al., 2022). For instance, Republicans tend to prefer the term *baby* over *fetus* to emphasize their belief that human rights begin at conception (Simon and Jerit, 2007), reflecting their

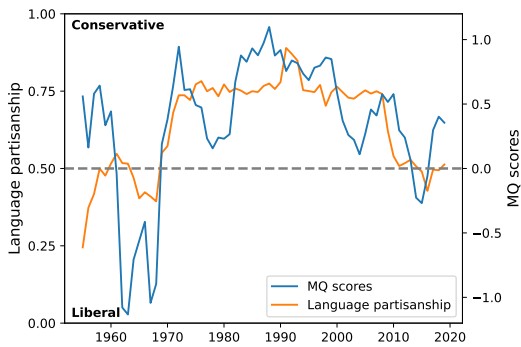

Figure 1: The US Surpreme Court overall partisanship between 1955 and 2019 as approximated by voting-based measures (MQ scores; Martin and Quinn (2007)) and our language-based framework.

beliefs and partisanship (Gentzkow et al., 2019; Demszky et al., 2019; Thomas et al., 2006; Bergam et al., 2022; Vafa et al., 2020).

In this paper, we ask *to what extent do oral arguments reveal justice partisanship*. We propose a framework to analyze partisanship in justices' oral arguments in the Supreme Court of the United States (SCOTUS). We classify justices as Democrat or Republican based on the political party of their nominating president, following common practice in the law literature (Devins and Baum, 2017; Yalof, 2001). We cast language-based partisanship prediction as a classification task and show that our models can reliably predict partisanship, suggesting that justices project their affiliations in court arguments.

We go on to derive language-based partisanship scores from our models, and ask *How does language-based partisanship align with established measures of voting-based ideology?* We show that our scores correlate well with established voting-based measures (Figure 1). Equipped with this layer of validation, we move on to more nuanced analyses of language-based partisanship: (a) of the overall court, (b) of individual justices, and

---

*Now at Google DeepMind.

[1]The dataset and code are available from: https://github.com/biaoyanf/SCOTUS-partisanship.

|  | N |  |
|---|---|---|
| Cases | 6,663 | |
| Justices | 35 | |
| Turns | 756,018 | |
| Average #words per turn | 27.24 | |
| Instances | 104,498 | |
|    Republican | 62,497 | (59.8%) |
|    Democratic | 42,001 | (40.2%) |

Table 1: Core dataset statistics, where "Turns" are individual justice contributions (of arbitrary length) and "Instances" are the individual data points presented to the models, i.e., 510-token segments with a sliding 255 token window from *all* turns of one justice in one case.

(c) of landmark cases over time.

## 2 Methodology

**Data** We use a subset of the Super-SCOTUS dataset (Fang et al., 2023b), which contains transcripts of SCOTUS oral arguments from 1955–2019 (Chang et al., 2020).[2] We filter out cases where hearings stretch over more than one year. We additionally remove non-linguistic indicators, e.g., *[laughter]*, and mask all person names with a BERT NER model[3] (Devlin et al., 2019) in order to focus our models on linguistic indicators rather than mentions of individuals with potential party affiliations.[4]

We concatenate all *turns* from an individual justice in a single court hearing, to derive multiple *instances* with a maximum of 510 tokens and a sliding window of 255 tokens. To retain only informative instances, we further remove those that have less than 50 tokens. We finally randomly separate the resulting instances into 10 folds for cross-validation. Partisanship and polarization are dynamic phenomena, responding to shifts in topics and political landscapes. We equip our models with a notion of time by adding a special [year] tag to each instance, flagging its year of origin.

We obtain the reference party affiliation for each justice as the party (Republican or Democrat) of the President who nominated the justice, which correlates strongly with the self-reported party of the justice and has been shown to be associated with the interest of the nomination party (Devins and Baum, 2017; Shipan, 2008; Yalof, 2001). The statistics of the resulting dataset are shown in Table 1.

| Model | Train | Dev | Test |
|---|---|---|---|
| Random | 0.50 ± 0.00 | 0.50 ± 0.00 | 0.50 ± 0.00 |
| Majority | 0.37 ± 0.00 | 0.37 ± 0.00 | 0.37 ± 0.00 |
| BERT | 0.87 ± 0.02 | 0.83 ± 0.01 | 0.83 ± 0.01 |
| BERT (+year) | 0.91 ± 0.01 | 0.85 ± 0.00 | 0.85 ± 0.01 |

Table 2: Macro-F1 scores for affiliated party prediction for Random and Majority baselines (majority=Republican, cf. Appendix B), as well as fine-tuned BERT classifiers based on justice instances only (BERT), or an additional year tag (BERT +year).

| Model | Train | Dev | Test |
|---|---|---|---|
| Random | 0.49 | 0.47 | 0.50 |
| Majority | 0.40 | 0.42 | 0.34 |
| BERT | 0.88 ± 0.02 | 0.80 ± 0.01 | 0.70 ± 0.01 |

Table 3: Macro-F1 scores for affiliated party prediction based on chronological data split. We show the averaged results of BERT over three runs with different random seeds.

**Classification** To understand to what extent their language reveals a justice's party affiliation (Democrat vs. Republican), we formulate partisanship prediction as a binary classification task, and fine-tune the BERT-base model[5] (Devlin et al., 2019) to predict the affiliated party of a justice from their oral contributions in a single court hearing. For each iteration of cross-validation, we fine-tune BERT on the training fold for 30 epochs, with a batch size of 16, and a dropout rate of 0.3, and select the best model based on the Macro-F1 score on the development fold.

## 3 Affiliated Party Prediction

We first evaluate our framework intrinsically, asking *To what extent do oral arguments reveal justices' party affiliation?* by predicting justices' affiliated party from their court contributions.

Table 2 shows affiliated party prediction performance of our classifiers.[6] The large boost for the BERT models over random and majority baselines, as well as the high overall performance, shows that the wording of the oral arguments reveals justices' party affiliation as a proxy for their partisanship. Furthermore, models benefit from temporal tags (+year), indicating that relevant linguistic signals drift over time.

To further investigate the impact of temporal lan-

[2]https://convokit.cornell.edu/documentation/supreme.html
[3]https://huggingface.co/dslim/bert-base-NER
[4]See Appendix A for data construction details.

[5]https://huggingface.co/bert-base-cased
[6]Distributions of gold vs. predicted labels over one fold are provided in Appendix B.

| | LR | SVM |
|---|---|---|
| BERT (off-the-shelf) | 0.45 ± 0.01 | 0.43 ± 0.01 |
| BERT (+ year) | 0.29 ± 0.06 | 0.32 ± 0.06 |

Table 4: Speaker identification performance (Macro-F1) of logistic regression (LR) and SVM classifiers, taking as input off-the-shelf BERT embeddings or embeddings from our best-performing party classification model.

guage shift (Hu et al., 2019; Ding et al., 2023) on our oral partisanship prediction task, we train and test our model with a chronological data split. We split the data into non-overlapping temporal spans, with the training covering all cases from 1955–2000 ($N = 5336$), development set (2001–2009, $N = 671$) and test set (2010–2019, $N = 656$). The results in Table 3 show that, compared to the random split (Table 2), BERT performs worse on the development and test sets. In line with prior work, this suggests that characteristics of partisan language change over time. Incorporating language drift into predictive models is a fertile area for future work.

A natural question is whether our model indeed captures partisanship, or rather language idiosyncrasies of individual justices. To test this, we use our trained models to predict speaker identities. Specifically, we extract the final layer embedding of the *[CLS]* token and train logistic regression (LR) and support vector machine (SVM) models for speaker prediction (i.e., a 35-way justice classification task). We compare against off-the-shelf BERT-base-cased embeddings, with no fine-tuning. We use instances with year tags as input, based on the best-performing setup in Table 2.[7] Our fine-tuned model performs worse than off-the-shelf BERT embeddings on the task of speaker prediction (Table 4). As such, we conclude that our fine-tuned model representations abstract away from individual speaker characteristics to representations that indeed capture properties indicative of party affiliation.

## 4 Justices' language reflects their voting

We next ask *How does language-based partisanship align with established measures of voting-based ideology?* To do so, we derive a partisanship score from our predicted party affiliation probabilities. We then relate our language-based ideology

scores to established measures of ideology obtained from justices' voting behavior (Martin and Quinn, 2002, 2007; Bonica and Sen, 2021). While partisanship and ideology are not identical, a strong correlation exists, particularly in the two-party system of the United States (Baum and Devins, 2019; Devins and Baum, 2017; Lupton et al., 2020).

Having shown that our BERT models capture partisanship from court arguments (Section 3) and with a grounded assumption that partisanship is reflected in (reasonably) local linguistic choice (Jarvis, 2004; Haddow and Klassen, 2006), we obtain a language partisanship score of a justice by averaging the predicted party probability for each of the justices' instances (Section 2). For example, to obtain the oral partisanship score of a given justice in a given year, we average the predicted party probability of all related instances from the given justice in the given year. Similarly, the language partisanship score for a case (year) is calculated by averaging the predicted party probability of all instances from all justices in that case (year), respectively.

We use MQ scores[8] (Martin and Quinn, 2002), a widely used and validated measure of the ideology of a court or individual justices, derived from voting outcomes. MQ scores are estimated with a dynamic Bayesian item response model (West and Harrison, 2006), which infers the latent 'ideal point' of a justice, i.e., their ideological standpoint on a unidimensional scale (Liberal — Conservative) based on their observed voting behavior and a prior encouraging a smooth change in ideology over time.

**Court-level partisanship**   Figure 1 compares the overall SCOTUS ideology based on voting behavior (MQ scores, orange) and language (blue) over time, from 1955 to 2019. We observe a strong correlation of both measures across time (Pearson's $r = 0.611$, $p = 6.5e{-}8$), indicating that partisanship in oral arguments reflects the voting behavior in court. Similar observations have been made for more overtly-partisan domains such as political discourse and votes in the US Congress (Gentzkow et al., 2019; Diermeier et al., 2012).

**Justice-level partisanship**   We further investigate language partisanship and voting ideology at the individual justice level. Justices in recent years have shown clearer alignment with party affilia-

---

[7]Appendix C shows that patterns are consistent for inputs without year tags.

[8]http://mqscores.lsa.umich.edu/measures.php

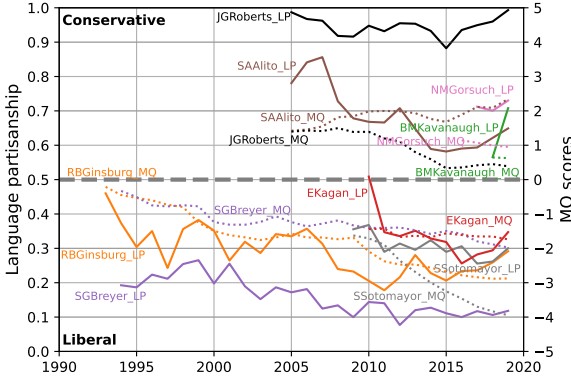

Figure 2: Comparison of voting-based (MQ; dashed lines) and language partisanship measures (LP; solid lines) of individual justices over their tenure. Different colors denote different justices. The bold-dashed line indicates the neutral point.

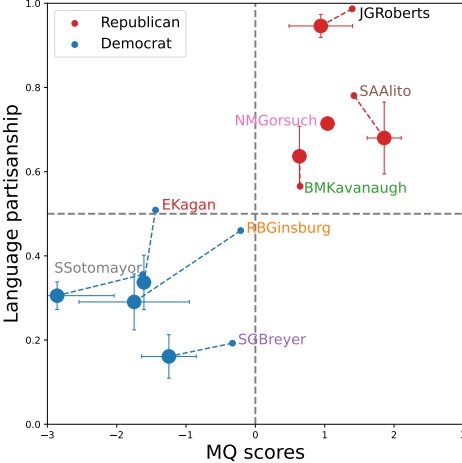

Figure 3: Drift of justice partisanship comparing their first year (small dot) and full tenure (large dot). Blue dots correspond to Democrat- and red dots to Republican-nominated justices. The error bars on large dots indicate the standard deviation of justice partisanship, i.e., MQ scores and language partisanship in corresponding axes, across their full tenure.

tions in terms of their voting patterns (Devins and Baum, 2017), and we ask whether this is reflected in their language, too. We present a case study of the eight most recently appointed justices in our data set.[9]

Figure 2 shows that Republican-affiliated/conservative justices (Roberts, Saalito, Gorsuch, Kavanaugh) are consistently separated by the neutral (thick, dashed) line from the Democrat/liberal justices (Ginsburg, Breyer, Kagan, Sotomayor). This holds for both language-based partisanship scores (solid lines) and MQ scores (dashed lines). For Democratic justices, MQ scores and language partisanship are aligned in tendency over their tenure, with a minimum of Pearson's $r = 0.6$ across justices. We do not observe such strong evidence in Republican justices, indicating that conservative justices project their values less directly in their speech. Language-based partisanship estimates tend to be more extreme than voting ideology. Notable examples are Roberts (Republican) and Breyer (Democrat), who is known to be a pragmatist whose decisions are often guided by real-life consequences regardless of party-lines,[10] possibly explaining the disparity between partisanship and MQ scores.

Epstein et al. (2007b) observed that, over their tenure, justices drift away from their first-year preferences, but with no certainty in what direction they will move. To study if the same observation holds

for language partisanship, we compare in Figure 3 how our eight most recent justices' voting ideology and language partisanship have drifted between their first year of tenure (small dots) vs. their entire tenure to date (large dots). The error bars indicate the standard deviation of partisanship scores (MQ on the $x$- and language-based on the $y$-axis) over all years of service. For most justices, the overall trend exceeds the SD interval, suggesting that it goes beyond the typical year-to-year fluctuation. As before, MQ and language scores align well (justices clustering in the bottom left and top right quadrants). Compared to their first year, Democrat-appointed justices (blue) statistically significantly tend to become *more liberal* in both MQ scores and language partisanship over their tenure. The relative shift in language partisanship for Republican-appointed justices (red) is less pronounced, in terms of both language and MQ scores.

**Case-level partisanship** In high-profile cases, voting behaviors have more distinctively lined up along party lines in the more recent years (Devins and Baum, 2017). Correspondingly, we might expect the gap between the average language partisanship in Democrat (blue) vs. Republican-affiliated (red) justices to increase over time. We test this by looking at 14 high-profile cases between 1962 and 2014 (Richard Wolf, 2015).[11] As shown in Fig-

---

[9]We excluded the most-recently appointed justice *Clarence Thomas* due to data sparsity.The same comparison for *all* justices is provided in Appendix D.

[10]https://www.oyez.org/justices/stephen_g_breyer

[11]Full details of the cases are provided in Appendix E.

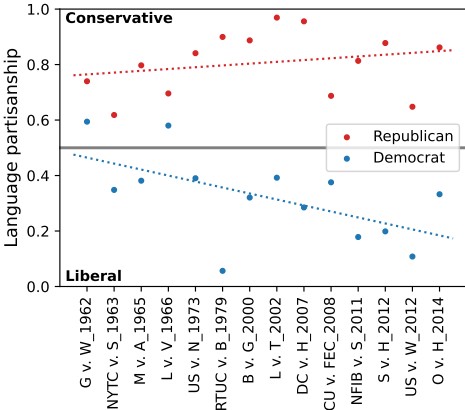

Figure 4: Language Partisanship on important cases. We average the language partisanship of justices that were nominated by the same party, i.e. Republican (red) or Democrat (blue). Dashed lines denote the estimated linear regression for the corresponding party.

ure 4, language partisanship indeed becomes more polarized. Particularly, the language of Democrats, which in the 1960s occasionally crossed the neutral line to the Conservative side, has become more liberal over time. Additionally, the gap in language partisanship between the two groups of justices has increased, again confirming that justices' final voting behavior is reflected in their spoken court arguments.

## 5 Discussion

Partisanship as affiliation along the liberal–conservative political spectrum is a fundamental axis in the political discourse. Automatically identifying partisan language through NLP techniques enables large-scale analysis of political and public discourse and a better understanding of divisions and polarization. While prior work has mostly focused on explicitly partisan text, e.g., congress speeches (Jensen et al., 2012) or news outlets (Dutta et al., 2022), we study supreme court language, which, by definition, should be politically neutral, and ask *does the language of justices in court reveal their party affiliation?* We proposed an analysis framework showing that BERT-based classifiers can reliably predict the affiliated party of justices based on linguistic signals in their SCOTUS arguments.

In line with research which has shown a correlation between conversational content and final voting in political discourse (Lupton et al., 2020; Vafa et al., 2020; Bonica and Sen, 2021; Bergam et al., 2022), we further asked *to what extent language partisanship aligns with voting ideology*. We derived language-based partisanship scores from our validated models and compared them with MQ scores, an established ideology measure derived from voting patterns. We showed a strong correlation between language partisanship and voting ideology at the overall court, individual justice, and important case level over time (Section 4), indicating that the oral arguments of justices do encode their political leanings. Moreover, our work reveals nuanced differences in linguistic and voting behaviors, e.g., their respective tendency shift over justices' tenure (Figure 3), which further demonstrates the importance of studying SCOTUS from various perspectives.

More broadly, we test the extent to which language representations from large language models capture nuanced socio-cultural phenomena, by comparing predictions against corresponding behavioral data sets. Here we focus on partisanship, building on related approaches for stance (Bergam et al., 2022) and counterfactual approaches to investigate the effect of social speaker attributes like gender and seniority on language use in the courtroom (Fang et al., 2023a).

Our proposed framework enriches the partisanship and ideology analysis with an additional dimension to voting behavior, the spoken text, where we show a strong correlation between voting and spoken language. We hope that this could further spur analyses from other dimensions, such as partisanship of advocates (Patton and Smith, 2017), and amicus curiae (Sim et al., 2016), and take important parts of the legal process, e.g., further opinion-writing (Clark, 2009), into consideration.

## Limitations

Our work focuses on the Supreme Court of the United States, due to a wealth of available resources and prior research to ground or results. Future work should extend our framework to other political corpora, e.g., congressional records (Gentzkow et al., 2019) and the federal circuit,[12] as well as languages and their associated political/legal systems.

We did not exhaustively search for the best language representation, and other language models, e.g. RoBERTa (Liu et al., 2019), XLNet (Yang et al., 2019), or BART (Lewis et al., 2020) may

---

[12]https://cafc.uscourts.gov/home/oral-argument/listen-to-oral-arguments/

lead to improved performance. Also, experiments with additional domain-specific language models, e.g., LEGAL-BERT (Chalkidis et al., 2020), and prompt engineering (Trautmann et al., 2022) may achieve a further increase in performance.

The temporal drift of language itself (e.g. different talking styles in court over time), as well as focus of topics (e.g. digital security only emerged in the past few decades), could be confounders in our partisanship analysis. Although we mitigate their impact by randomly splitting instances per year across folds, we acknowledge that these confounders are worthy of further exploration. Our use case suggests itself as a testbed for causal modeling approaches that control for confounders explicitly (Feder et al., 2022).

We acknowledge that it's hard to obtain reliable features, i.e., important words that impact prediction results, out of large language models. Our analysis framework could be further enhanced by inspecting the highly-associated words for the parties.

## Ethics Statement

Our models predict party affiliation and explicitly *not* voting outcomes. We neither attempt nor are able to do voting prediction based on our analysis framework, noting the severe ethical concerns it would raise (including, but not limited to, associating individuals with predicted professional behavior of potentially low quality, that may reflect detrimentally on their reputation).

Our analysis framework aims to understand how oral court arguments reveal the affiliated party of justices in the Supreme Court of the US. Although our study covers the analysis of individual justices, we make no presumptions of the values or beliefs of justices beyond what is in the public domain, nor do we target individual justices. All our analyses aggregate predictions over several cases per justice and we do not aim to predict or overly rely on individual votes or contributions of individuals. This research aids in understanding the US legal system, and we strongly advise against over-interpretation of the results in terms of behaviors of individual justices.

Our case study focuses on a subset of historical oral arguments from SCOTUS. Although it covers most publicly-available cases, this does not reflect the full history of SCOTUS, nor represent the current state of the court.

It has been shown that pretrained language models are biased (Delobelle et al., 2022; Nadeem et al., 2021). Although our evaluation validates our results against external sources, we acknowledge a possible impact of biases in language representations (e.g., resulting from a prevalence of liberal or conservative sources in the pre-training corpora).

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

## A  Data Construction

We removed all nonlinguistic information from the transcripts, including indicators of cross-talk (e.g. [voice overlap], [interruption]), nonverbal expressions (e.g. [laughter], [sighs], [applause]), and procedural markers (e.g. [luncheon], [recess]). The full list is made available as part of the code repository.

## B  Distribution of Gold and Predicted Affiliated Party

Figures 5 shows the label distribution of gold vs. predicted for affiliated party prediction over one randomly-selected fold.

## C  Detailed Experiment on Justice Identification Prediction

Table 5 provides a detailed experiment on justice identification prediction with and without year tags on instances.

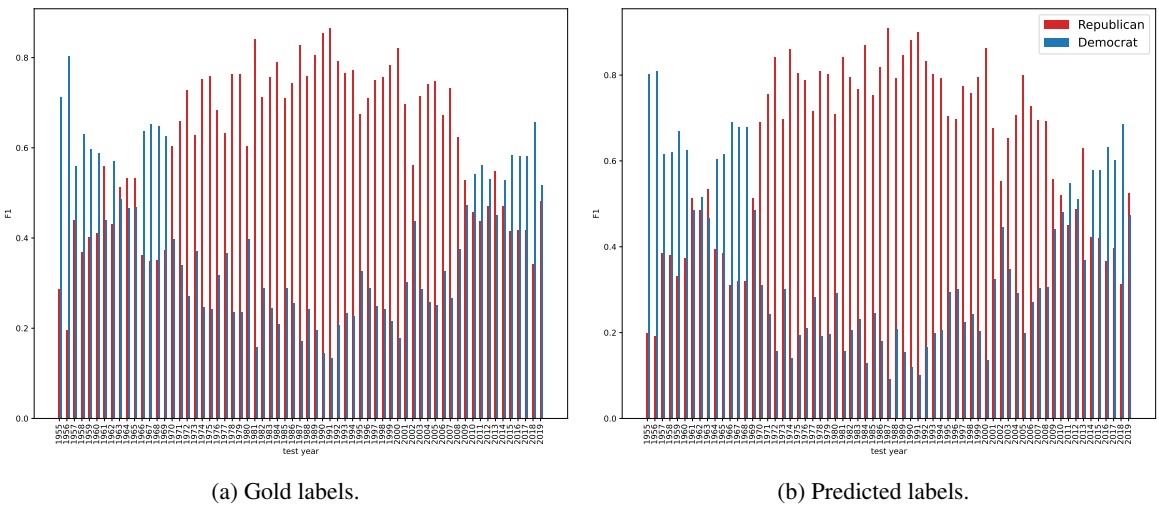

(a) Gold labels.                (b) Predicted labels.

Figure 5: Label distribution of gold vs. predicted for affiliated party prediction in our data set (as exemplified by one representative test fold).

|  | LR | SVM |
| --- | --- | --- |
| **Instances - year** | | |
| BERT (Vanilla) | 0.44 ± 0.01 | 0.43 ± 0.01 |
| BERT (Fine-tuned) | 0.32 ± 0.10 | 0.35 ± 0.10 |
| **Instances + year** | | |
| BERT (Vanilla) | 0.45 ± 0.01 | 0.43 ± 0.01 |
| BERT (Fine-tuned) | 0.29 ± 0.06 | 0.32 ± 0.06 |

Table 5: Macro-F1 scores for justice identity prediction over instances with year tags on 10-fold cross-validation. LR and SVM were trained on the training fold and performances were reported on the test folds.

## D  Comparison of Voting-based and Language-based Partisanship at Individual Justice Level

Figure 6 shows the comparison of voting-based (MQ scores; Martin and Quinn (2007) and language partisanship measures of all individual justices over their tenure.

Figure 7 shows the Comparison of all justice partisanship in their first year and over their tenure.

## E  High Profile SCOTUS Cases

Table 6 lists the details of the high-profile SCOTUS cases analyzed in Section 4, Figure 4.

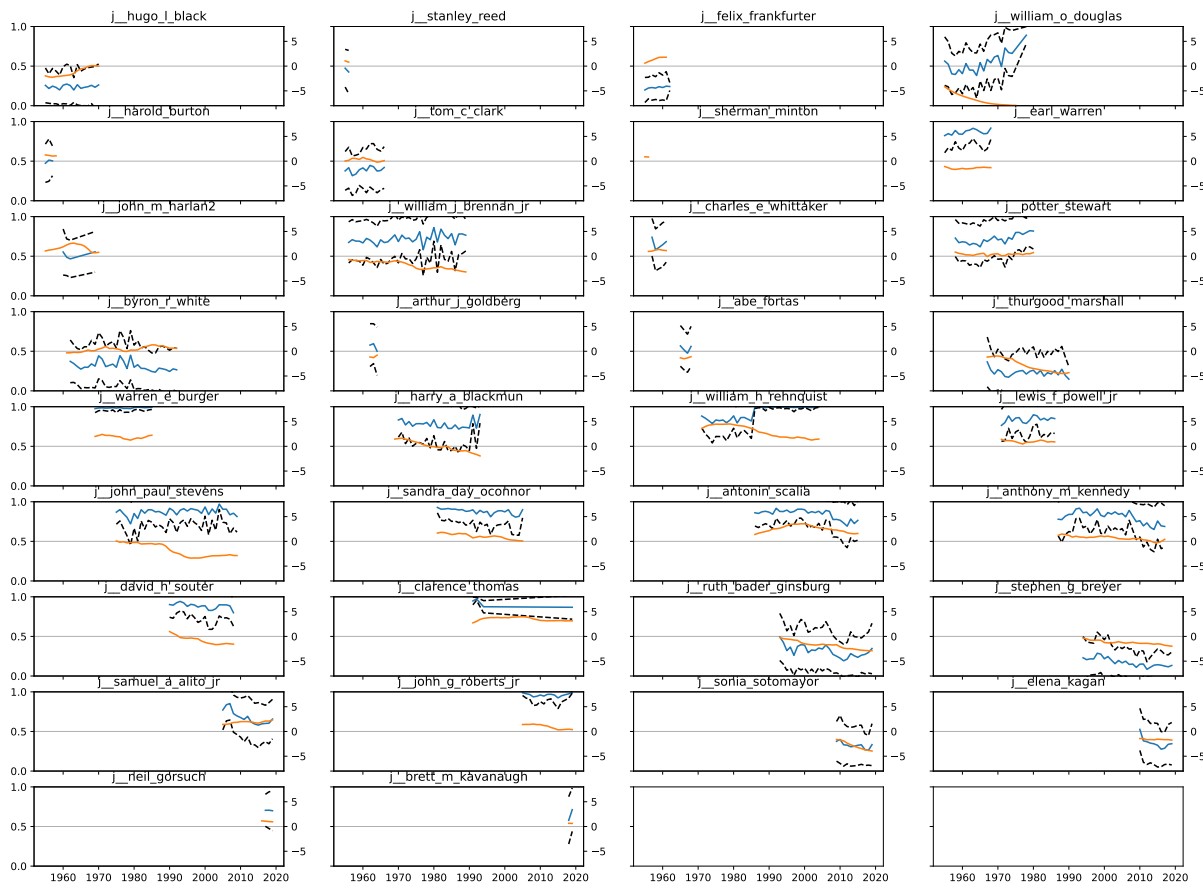

Figure 6: Comparison of voting-based (MQ scores (Martin and Quinn, 2007) (orange lines) and language partisanship measures (blue lines) of individual justices over their tenure. The dashed line notes the standard derivation of language-based partisanship measures.

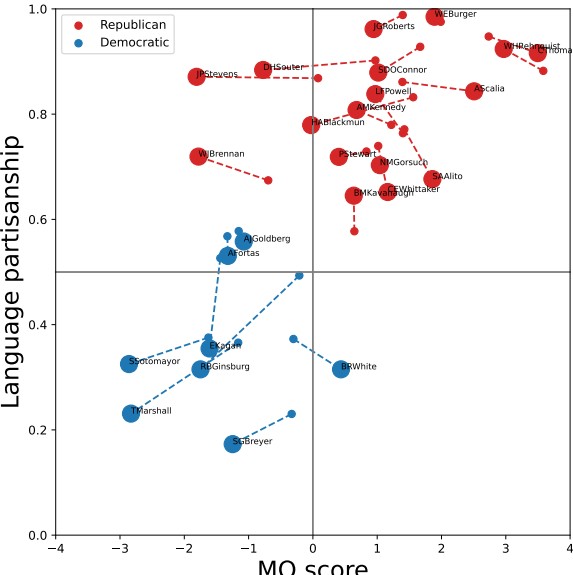

Figure 7: Comparison of justice partisanship in their first year (small dots) and over their tenure (big dots). Blue and Red dots denote justices that are affiliated with the Democrat and Republican parties, respectively.

| Year | Title | Issue Area | Abbreviated Label |
|------|-------|-----------|-------------------|
| 1962 | Gideon v. Wainwright | Criminal Procedure | G v. W_1962 |
| 1963 | New York Times Company v. Sullivan | First Amendment | NYTC v. S_1963 |
| 1965 | Miranda v. Arizona | Criminal Procedure | M v. A_1965 |
| 1966 | Loving v. Virginia | Civil Rights | L v. V_1966 |
| 1973 | United States v. Nixon | Criminal Procedure | US v. N_1973 |
| 1979 | Regents of the University of California v. Bakke | Civil Rights | RTUC v. B_1979 |
| 2000 | Bush v. Gore | Civil Rights | B v. G_2000 |
| 2002 | Lawrence v. Texas | Privacy | L v. T_2002 |
| 2007 | District of Columbia v. Heller | Criminal Procedure | DC v. H_2007 |
| 2008 | Citizens United v. Federal Election Commission | First Amendment | CU v. FEC_2008 |
| 2011 | National Federation of Independent Business v. Sebelius | Federalism | NFIB v. S_2011 |
| 2012 | Shelby County v. Holder | Civil Rights | S v. H_2012 |
| 2012 | United States v. Windsor | Due Process | US v. W_2012 |
| 2014 | Obergefell v. Hodges | Due Process | O v. H_2014 |

Table 6: List of high-profile cases in SCOTUS (Richard Wolf, 2015).