# OpenReview forum: "More than Votes? Voting and Language based Partisanship in the US Supreme Court"
_EMNLP/2023/Conference — EMNLP 2023 Findings_

### Official Review · Reviewer_NGhd · 2023-08-03

**Typos Grammar Style And Presentation Improvements:** Nicely written paper.
**Soundness:** 4

**Excitement:**

3: Ambivalent: It has merits (e.g., it reports state-of-the-art results, the idea is nice), but there are key weaknesses (e.g., it describes incremental work), and it can significantly benefit from another round of revision. However, I won't object to accepting it if my co-reviewers champion it.

**Paper Topic And Main Contributions:**

- The paper measures partisanship based on the text supreme court justices say during oral argument (and not the text in the written opinions).  This is interesting since it measures the text of individual justice and not the written opinions (that are joint statements).  The paper uses the existing idea that a model that regresses party affiliation (known) on text is informative.  The coefficients of that regression tell you what words correlate with party affiliation.  Here it is not literally a text regression but using the BERT structure.

**Questions For The Authors:**

A:  A sentence or two on how you derive "a partisanship score from our predicted party affiliation probabilities" would be helpful.  In particular, how that works per speaker per year?  I think this is mentioned in:  "we obtain a language partisanship score of a justice by averaging the predicted party probability for each of the justices’ instances."   Is the model predicting party or party probability?  (I think this is a familiar caution about interpreting logistic-style regressions as likelihoods).

B: How does the model deal with different volumes of text per speaker?  Some justices ask more questions.

C: Are there any insights into the language from the model that are interesting?  The simpler text-regressions have the advantage of being easy to interpret  ("baby" versus "fetus" is mentioned in the text).  Can this be captured here?

D:  This is beyond the scope of the paper, but perhaps worth asking:  Does the questions/text in oral argument predict the voting outcome?

E: Figure 3. It is interesting to compare first year to the whole tenure.  However, it would helpful to have a notion of "standard errors" here.  Is the movement from year one "big" or is this just movement year to year?

**Reasons To Accept:**

- This paper is clear, straightforward, and insightful.  Using the text from oral arguments is a good idea and generates interesting results.
- The analysis of individual justices (over time; relative to the first year on the court) is insightful

**Reasons To Reject:**

- This is a topic that has been heavily studied.  (And, to be clear, the paper notes this with thoughtful and concise citations).  So generating novel results here is more difficult.  What do we learn here that is new?


**Reproducibility:**

4: Could mostly reproduce the results, but there may be some variation because of sample variance or minor variations in their interpretation of the protocol or method.

**Reviewer Confidence:**

3: Pretty sure, but there's a chance I missed something. Although I have a good feel for this area in general, I did not carefully check the paper's details, e.g., the math, experimental design, or novelty.

---

> ### Author Rebuttal · Authors · 2023-08-29
>
> Thank you for your valuable feedback and questions.
>
> **Significance of contribution.** The importance of the partisanship analysis in our paper can be summarized into three areas: (1) Prior work on partisan language classification in NLP has mostly focused on explicitly partisan text (such as congress speeches or news outlets), while prior work on bias in SCOTUS has largely ignored the court arguments. We study supreme court language, which, by definition, should be politically neutral. Consequently, the fact that we can reliably predict party affiliation from court language is a novel and noteworthy finding in itself. (2) We link the language in oral arguments to justice voting behavior in the SCOTUS. Our analysis shows for the first time that voting partisanship is reflected in court language. And (3) our work reveals nuanced differences in linguistic and voting behaviors e.g. the tendency shift in Fig 3 (First year vs. Tenure), which further demonstrates the importance of studying SCOTUS from various perspectives.  We will clarify these points in the paper.
>
>
> To answer your questions:
>
> Question A. For each data point (i.e., contribution by a given justice in a given year), the model predicts the probability p(y=Democrat). We average the predicted probability e.g., to obtain a justice-level score.  In preliminary experiments, we also experimented with aggregating class predictions (rather than probabilities), observing similar results. We will clarify this in the paper.
>
>
>
> Question B. To ensure sufficient data for each justice in our analysis, we excluded justices with fewer than five data points for every single year in our data set. Additionally, by using a sliding window we leverage the full available data per justice.
>
>
>
> Questions C. This is a great suggestion, but as discussed in the Limitations section, we emphasize the difficulty in obtaining reliable salient words from large language models. A concerted attempt is beyond the scope of this (short) paper.
>
>
>
> Question D. Thanks for the suggestion. First, we would like to point out the ethical issues raised by applications that predict voting behavior based on language. That said, we can confirm from preliminary experiments that voting prediction based on our representation was poor. One hypothesis is that the spoken language does not give away the voting information as justices often make up their mind only after the full argument (in fact, we would hope this is the norm). So, the justice language in the court room might reflect their (static) party affiliation, but the oral argument itself does not reveal their voting behaviour. Other official documents, such as Court Opinions, capture this stage.
>
>
>
> Question E. We will add the analysis of “standard errors”, i.e., average year-to-year movement for justices, to Fig 3.

---

### Official Review · Reviewer_2Gwe · 2023-08-04

**Soundness:** 4

**Excitement:**

3: Ambivalent: It has merits (e.g., it reports state-of-the-art results, the idea is nice), but there are key weaknesses (e.g., it describes incremental work), and it can significantly benefit from another round of revision. However, I won't object to accepting it if my co-reviewers champion it.

**Paper Topic And Main Contributions:**

The paper examines partisan language in SCOTUS oral arguments. The authors attempt to answer two research questions: 1.) Does a justice's speech in oral arguments indicate their political affiliation? 2.) How well do oral arguments correlate to other partisanship measures?

The main contribution of this paper is a speech-based alternative to voting-based partisanship measures, specifically, a classification model to predict party affiliation based on utterances.

**Questions For The Authors:**

A. Why is partisanship measure important to political science research?
B. You noted word choice conveys specific messages in the introduction. Wouldn't model interpretability be a vital aspect of this approach?   If so, have you analyzed the salient words or phrases?

**Reasons To Accept:**

The paper provides a novel proxy for partisanship based on speech instead of voting records. The paper's main strengths include:
- Well-documented data prep and model training procedures.
- Analysis to ensure the model isn't focused on speaker identification
- Temporal analysis of language partnership and MQ scores.

**Reasons To Reject:**

The paper must clearly state why the partisanship/ideology measure is essential or how this measure impacts political science research.   The approach would be better if it provided insights into linguistic features that indicate partisanship in the utterances.

**Reproducibility:**

4: Could mostly reproduce the results, but there may be some variation because of sample variance or minor variations in their interpretation of the protocol or method.

**Reviewer Confidence:**

4: Quite sure. I tried to check the important points carefully. It's unlikely, though conceivable, that I missed something that should affect my ratings.

---

> ### Author Rebuttal · Authors · 2023-08-29
>
> Thank you for your comments.
>
> **Significance of contribution.** The importance of the partisanship analysis in our paper can be summarized into two folds: (1) We link the language in oral arguments to justice voting behavior in the SCOTUS. Our analysis shows for the first time that voting partisanship is reflected in court language. Moreover, our work reveals nuanced differences in linguistic and voting behaviors e.g. the tendency shift in Fig 3 (First year vs. Tenure), which further demonstrates the importance of studying SCOTUS from various perspectives.  (2) More fundamentally, partisanship as affiliation along the liberal---conservative political spectrum is a fundamental axis in the (western) political discourse. Automatically identifying partisan language through NLP techniques enables large-scale analysis of political and public discourse, and better understanding of divisions and polarization. Our use case is one example of this. We will add this clarification to the paper.
>
> **Linguistic features.** We address the issue of feature analysis in the Limitations section. While we agree that analyzing partisanship-indicators is a valuable direction for future work, it is outside the scope of this paper. We acknowledge the difficulty of obtaining reliable linguistic features from large language models --- particularly given the substantial time span covered in our data. Obtaining salient words from oral arguments is a promising future venture to enhance the analysis of oral partisanship.

---

### Official Review · Reviewer_HTaT · 2023-08-09

**Typos Grammar Style And Presentation Improvements:** I found no typos, and the presentatio…
**Soundness:** 3

**Excitement:**

3: Ambivalent: It has merits (e.g., it reports state-of-the-art results, the idea is nice), but there are key weaknesses (e.g., it describes incremental work), and it can significantly benefit from another round of revision. However, I won't object to accepting it if my co-reviewers champion it.

**Missing References:**

This is one of the earlier works that study partisan language in political discourse.

1. Jensen, Jacob, et al. "Political polarization and the dynamics of political language: Evidence from 130 years of partisan speech [with comments and discussion]." Brookings Papers on Economic Activity (2012): 1-81.

The following paper showed that it is possible to predict the source of a text in a particular domain based on the partisan linguistic signals present in the news transcripts.

2. Dutta, Sujan, et al. "A Murder and Protests, the Capitol Riot, and the Chauvin Trial: Estimating Disparate News Media Stance." Proceedings of the Thirty-First International Joint Conference on Artificial Intelligence, IJCAI-22. 2022.





**Paper Topic And Main Contributions:**

This paper proposes a framework for analyzing the language of justices (during oral arguments) to identify partisan signals, demonstrating the predictability of justices' affiliated parties based on their oral contributions. The authors found a strong correlation between the justices' language partisanship and their voting behavior.

**Reasons To Accept:**

Analyzing the language-based partisanship of supreme court justices is interesting. The paper compares the affiliated party prediction model's performance on speaker identity detection against an off-the-shelf model to conclude that the model is not capturing the idiosyncracies of individual justices. Then it correlates the partisanship score with the voting ideology (MQ score). The work also conducts deeper analysis at the individual justice level and case level. Overall, this work broadens the understanding of partisan language in oral arguments used by the justices.

**Reasons To Reject:**

The findings of this paper are somewhat expected. Existing work has studied linguistic differences among (groups of) people with different political ideologies.

**Reproducibility:**

5: Could easily reproduce the results.

**Reviewer Confidence:**

4: Quite sure. I tried to check the important points carefully. It's unlikely, though conceivable, that I missed something that should affect my ratings.

---

> ### Author Rebuttal · Authors · 2023-08-29
>
> Thank you for your feedback.
>
> We agree that prior work has demonstrated that ideology can be predicted from texts that are expected to capture author stance (such as congress speeches in Jensen et al., (2012) or practically party-affiliated US news channels in Dutta et al., (2021)). We study supreme court arguments which, by definition, should be politically neutral. Consequently, the fact that we can reliably predict party affiliation from court language is a noteworthy finding. The fact that our language-based predictions correlate overall with established voting-based measures of justice partisanship is reassuring and validates our method. Moreover, our method enables deeper insights, for instance by studying the varying degree of discrepancy between words (language) and action (voting) for individual justices (Fig. 2). We will clarify this point in the paper.

---

### Official Review · Reviewer_9L3i · 2023-08-09

**Soundness:** 4

**Excitement:**

3: Ambivalent: It has merits (e.g., it reports state-of-the-art results, the idea is nice), but there are key weaknesses (e.g., it describes incremental work), and it can significantly benefit from another round of revision. However, I won't object to accepting it if my co-reviewers champion it.

**Missing References:**

1. **Consider the comprehensiveness and comparison of the models.**

I recommend:

Trautmann, D., Petrova, A., & Schilder, F. (2022). Legal prompt engineering for multilingual legal judgement prediction. arXiv preprint arXiv:2212.02199.

2. **Consider diachronic word embeddings and semantic shifts to make the research more comprehensive**

I recommend:

Ding, X., Horning, M., & Rho, E. H. (2023, June). Same Words, Different Meanings: Semantic Polarization in Broadcast Media Language Forecasts Polarity in Online Public Discourse. In Proceedings of the International AAAI Conference on Web and Social Media (Vol. 17, pp. 161-172).

Hu, R., Li, S., & Liang, S. (2019, July). Diachronic sense modeling with deep contextualized word embeddings: An ecological view. In Proceedings of the 57th Annual Meeting of the Association for Computational Linguistics (pp. 3899-3908).

**Paper Topic And Main Contributions:**

The paper addresses a significant gap in the understanding of partisanship and ideology in the US Supreme Court. Building on the many previous studies that have quantified partisanship based on judicial voting patterns and certain courtroom behaviors, the authors present a comprehensive approach to address the existing research gaps concerning the content of oral arguments in courtrooms. Specifically, this paper seeks to understand:

- To what extent do the oral arguments made by justices in the US Supreme Court reflect their partisan or ideological beliefs?

- Can linguistic patterns in these oral arguments be used to predict the affiliated party or ideology of the justices?

- How does language-based partisanship align with established measures of voting-based ideology?

**Contributions**

*Highlighting the Significance of Oral Arguments*. The paper underscores the importance of oral arguments in the courtroom as a potential measure of partisanship. It emphasizes that while voting patterns are a direct measure, the nuances in the language during oral arguments can offer deeper insights into their beliefs and partisanship.

*Bridging NLP technology and oral argument in SCOTUS*. The paper leverages the BERT-based classifiers which can be employed effectively to predict the affiliated party of justices. And the authors' work indicates a successful marriage of legal studies/political corpora and advanced computational techniques, showcasing the potential of NLP for a wider range of research areas.

*Correlation Analysis of Behavior and Linguistic Patterns*. The authors correlate language-based ideology scores with well-established ideology measures obtained from justices' voting patterns. By building on previous research, which primarily quantifies partisanship through voting behavior, this paper further corroborates and analyzes the connection between voting and spoken language.

**Reasons To Accept:**

**Clear Structure**

The overall methodology and research pipeline of the paper are well-defined. Firstly, the research motivation logically stems from prior studies, which primarily quantify partisanship based on voting behavior and oral arguments in the courtroom. The research method predominantly employs BERT as the foundational model for classification, adeptly capturing the binary classification results. Lastly, to substantiate the significance of the results, an analysis of how the Justices' language reflects their voting was undertaken.

**Effective Correlation Analysis**

From the focal point of the research, this paper emphasizes *"How does language-based partisanship align with established measures of voting-based ideology"*. The paper employs enough graphs and tables to substantiate the correlation between voting behavior and spoken language. Moreover, the approach of comparing voting-based and language partisanship measures of individual justices throughout their tenure is methodologically sound.

**Insightful Perspective**

The authors' choice to analyze the alignment from a temporal standpoint is commendable and holds significant value. This perspective offers fresh insights and paves the way for future studies that converge political systems with linguistic inquiries.

**Reasons To Reject:**

**Research Integrity:**

From a data perspective, the SCOTUS dataset spans from 1955 to 2019. Undoubtedly, this represents a significant time frame, which implies that the authors aim to classify from a more comprehensive temporal standpoint. However, this extended time span also introduces variations in linguistic styles. For a binary classification (Democrat vs. Republican), the linguistic evolution within each party over several decades can pose challenges. Specifically, the choice of vocabulary and linguistic expressions for both the Democrats and Republicans have likely evolved over this period, which implies that achieving a thorough understanding and classification can be challenging.

 *Therefore, the lack of analysis of the temporal dimension in relation to linguistic features/patterns is not sufficient to demonstrate the accuracy and significance of the classification model.*

1. The study could venture into examining the linguistic evolution within each party. Taking the Democrats as an example, the authors could use Sentence-BERT or other large language models to extract sentence embeddings. They could then calculate the similarity between embeddings across years (e.g., using cosine similarity, etc.) to gauge if there has been an "evolutionary" shift in the Democrats' linguistic patterns from 1955 to 2019.
2. Building on this, the similarity between embeddings could be incorporated as an additional feature in the classification process, bolstering the robustness of their analysis.

**Baseline Work:**

In this study, the authors fine-tuned the BERT model, experimented with adding temporal tags [year], and tried embedding the last layer of BERT into traditional ML models. While these represent innovative attempts, it's better to benchmark against more baseline models or those from relevant domains. This not only showcases the novelty of the authors' approach but also positions their contributions in the larger context. Models like LegalBERT, CaseLawBERT, or LegalRoBERTa could serve as relevant benchmarks. Furthermore, I recommend that the authors consider models like the Longformer. Given its maximum input sequence length of 2,048 tokens, it might enhance the model's ability to perceive broader contextual relationships.

*Therefore, the lack of sufficient and comprehensive model comparisons, and the limitation of the length of the text inputs, affect the comprehensiveness of the results and the creativity of the author's methodology.*

**Reproducibility:**

4: Could mostly reproduce the results, but there may be some variation because of sample variance or minor variations in their interpretation of the protocol or method.

**Reviewer Confidence:**

4: Quite sure. I tried to check the important points carefully. It's unlikely, though conceivable, that I missed something that should affect my ratings.

---

> ### Author Rebuttal · Authors · 2023-08-29
>
> Thank you for your valuable feedback.
>
> 1. **Temporal language shift.** We agree with your concern about language shift over the time period of our data, and we discuss this in the Limitations section. Our work shows that it is possible to capture partisan language in this data set without an explicit model of time: we split our data into 10 folds ensuring coverage of the full period in each fold. An explicit representation of language shift may well improve performance and is a great avenue for future work. In preliminary experiments, we quantified the impact of language drift on performance by experimenting with a chronological data split (the test set covering strictly later years than the training set). We will add this analysis to the paper.
>
> 2. **Benchmark models.** This paper asks, “To what extent do oral arguments reveal justice partisanship”. We use the BERT models as a benchmark to explore if it is possible to infer justice partisanship based on their oral arguments, and present positive results. In our experiments, we include random/majority prediction baselines, and we observe a large improvement with BERT.   Experiments with additional domain-specific language models may very well show a slight increase of performance but this is not the focus of this (short) paper, and not necessary to answer our research question.
>
> 3. **Input length.** We evaded the input limitation of the BERT models by using overlapping sliding windows with shifts of 255 tokens. In Section 4, we consider the average partisanship value across all data points (windows) per justice in each year. We effectively assume that partisanship is reflected in (reasonably) local linguistic choice. This assumption builds on prior work which showed that partisanship is reflected in word choice (see e.g., Jarvis (2004); Robinson et al (2007), cited in the paper). We will clarify our assumptions.
>
> 4. We will add the suggested references to the paper.

---

### Meta-Review · Area_Chair_HPH6 · 2023-09-18

**Recommendation:** 3

**Metareview:**

Three reviewers evaluated the paper’s soundness with 4 (strong) and one with 3 (good) – overall, indicating that the paper on the most part provides sufficient support for all of its claims/arguments.  All four reviewers, however, were ambivalent about the paper’s excitement (3) – noting some ambiguity in the novelty of the contribution and potential limitations. Reviewer 9L3i  raised concerns about the limitation of not considering the diachronic aspects and more comprehensive model comparisons. Reviewer 2Gwe requested the authors to clarify the importance of their partisanship measure. 2Gwe also noted, the paper would be strengthened by providing key insights into linguistic features that indicate partisanship in the justice languages. Reviewer NGhD also notes that this is a topic that has been heavily studied and argues that the paper would benefit from highlighting new contributions. Relatedly, Reviewer HTaT notes that existing work has studied linguistic differences among (groups of) people with different political ideologies. An additional analysis examining the partisan differences in the language of justices would address such reviewers’ concerns. While the authors addressed some limitations with relevant citations and provided clarification to the reviewers’ questions, the paper will still benefit from additional analyses suggested by 2Gwe and 9L3i.

---

### Decision · Program_Chairs · 2023-10-07

**Decision:**

Accept-Findings

**Comment:**

Three reviewers evaluated the paper’s soundness with 4 (strong) and one with 3 (good) – overall, indicating that the paper on the most part provides sufficient support for all of its claims/arguments.  All four reviewers, however, were ambivalent about the paper’s excitement (3) – noting some ambiguity in the novelty of the contribution and potential limitations. Reviewer 9L3i  raised concerns about the limitation of not considering the diachronic aspects and more comprehensive model comparisons. Reviewer 2Gwe requested the authors to clarify the importance of their partisanship measure. 2Gwe also noted, the paper would be strengthened by providing key insights into linguistic features that indicate partisanship in the justice languages. Reviewer NGhD also notes that this is a topic that has been heavily studied and argues that the paper would benefit from highlighting new contributions. Relatedly, Reviewer HTaT notes that existing work has studied linguistic differences among (groups of) people with different political ideologies. An additional analysis examining the partisan differences in the language of justices would address such reviewers’ concerns. While the authors addressed some limitations with relevant citations and provided clarification to the reviewers’ questions, the paper will still benefit from additional analyses suggested by 2Gwe and 9L3i.